

# Sexual behaviour and risk of sexually transmitted infections in young female healthcare students in Spain

Felipe Navarro-Cremades[1], Antonio Palazón-Bru[1], Dolores Marhuenda-Amorós[2], María Isabel Tomás-Rodríguez[2], Fina Antón-Ruiz[3], Josefina Belda-Ibañez[4], Ángel Luis Montejo[5] and Vicente Francisco Gil-Guillén[1]

[1] Department of Clinical Medicine, Miguel Hernández University, San Juan de Alicante, Alicante, Spain
[2] Department of Pathology and Surgery, Miguel Hernández University, San Juan de Alicante, Alicante, Spain
[3] Department of Education, San Antonio Catholic University, Murcia, Spain
[4] Centre for Information and AIDS Prevention, Conselleria de Sanitat, Alicante, Spain
[5] Department of Nursing, University of Salamanca, Salamanca, Spain

Corresponding author
Antonio Palazón-Bru,
antonio.pb23@gmail.com

## ABSTRACT

**Background.** Several authors have examined the risk for sexually transmitted infections (STI), but no study has yet analyzed it solely in relation with sexual behaviour in women. We analyzed the association of sexual behaviour with STI risk in female university students of healthcare sciences.

**Methods.** We designed a cross-sectional study assessing over three months vaginal intercourse with a man. The study involved 175 female university students, without a stable partner, studying healthcare sciences in Spain. Main outcome variable: STI risk (not always using male condoms). Secondary variables: sexual behaviour, method of orgasm, desire to increase the frequency of sexual relations, desire to have more variety in sexual relations, frequency of sexual intercourse with the partner, and age. The information was collected with an original questionnaire. A logistic regression model was used to estimate the adjusted odds ratios (ORs) in order to analyze the association between the STI risk and the study variables.

**Results.** Of the 175 women, 52 were positive for STI risk (29.7%, 95% CI [22.9–36.5%]). Factors significantly associated with STI risk ($p < 0.05$) included: orgasm (not having orgasms → OR = 7.01, 95% CI [1.49–33.00]; several methods → OR = 0.77, 95% CI [0.31–1.90]; one single method → OR = 1; $p = 0.008$) and desiring an increased frequency of sexual activities (OR = 0.27, 95% CI [0.13–0.59], $p < 0.001$).

**Conclusions.** Women's desire for sexual activities and their sexual function were significant predictors of their risk for STI. Information about sexual function is an intrinsic aspect of sexual behaviour and should be taken into consideration when seeking approaches to reduce risks for STI.

Subjects Epidemiology, HIV, Infectious Diseases, Psychiatry and Psychology, Public Health
Keywords Condoms, Sexual behaviour, Sexual partners, Women, Sexually transmitted diseases

## INTRODUCTION

Sexually transmitted infections (STI) are spread primarily through person-to-person sexual contact. There are more than 30 different sexually transmissible bacteria, viruses

and parasites (*World Health Organization, 2015*). The most common conditions caused are gonorrhea, chlamydial infection, syphilis, trichomoniasis, chancroid, genital herpes, genital warts, human immunodeficiency virus (HIV) infection and hepatitis B infection (*World Health Organization, 2015*). The latex condom for males is the single most efficient available technology to reduce the sexual transmission of HIV and other STI as well as offering protection for the prevention of unintended pregnancy (*Holmes, Levine & Weaver, 2004*). For example, the risk of a woman contracting HIV per unprotected sexual act with an infected man is 0.1% (twenty times less per protected sexual act) (*Varghese et al., 2002*).

In Spain, about one in every seven women under the age of 30 years has had sexual intercourse with a casual partner during the previous year (*Instituto Nacional de Estadística, 2004*). Moreover, as only three in five always use a condom with casual partners, an important percentage of these women are at risk of contracting a STI (*Instituto Nacional de Estadística, 2004*). The percentage of women who are married is very similar in Spain to the rest of Europe, as well as to the US, though it is difficult to compare with rates in Africa as these data are not always recorded (*Orten, 2008*; *Instituto Nacional de Estadística, 2011*; *Herbenick et al., 2013*). STI, however, are more common in developed countries (Spain, US, etc.), followed by south and southeast Asia and sub-Saharan Africa (*Gewirtzman et al., 2011*). The most prevalent STI in women are trichomoniasis (Spain and Africa), HPV (US) and chlamydia (Europe) (*World Health Organization, 2001*; *Instituto Nacional de Estadística, 2004*; *Johnson, Coetzee & Dorrington, 2005*; *Gewirtzman et al., 2011*; *European Centre for Disease Prevention and Control, 2010*; *Satterwhite et al., 2013*). Spanish women, though, are less unfaithful than others in Europe, or in Africa and the US (*Conley et al., 2012*; *The Institut français d'opinion publique, 2014*; *Onoya et al., 2015*). In Spain, as everywhere, religion is a factor that influences the use of condoms (*Cooksey, Rindfuss & Guilkey, 1996*; *Instituto Nacional de Estadística, 1999*; *Lazarus et al., 2009*; *De Neve, 2013*).

Scholars have examined risky sexual behaviour in different populations but none have focused specifically on sexual behaviour and sexual function in women (Table 1). Studies have been done, though, on the association between sexual behaviour and infection with a particular STI (not the risk of contracting the infection), such as bacterial vaginosis (*Nilsson et al., 1997*), or the association between the use of a condom at sexual debut and the STI risk in American adolescents (*Shafii, Stovel & Holmes, 2007*). To fill this gap in the literature, we designed a study involving university women studying for a healthcare qualification. Questionnaires assessed the lack of use of the male condom during sexual activities with casual partners and its association with various different types of sexual function/behaviour. The results of the study suggest the need for future educational measures to prevent STI in healthcare personnel.

## MATERIALS & METHODS

### Study population

The study involved women studying for a healthcare qualification at Miguel Hernández University (Medicine, Pharmacy, Physiotherapy, Podiatry, and Occupational Therapy) in San Juan de Alicante (Spain). These degrees are studied by 28.3% of Spanish women

**Table 1   Studies evaluating sexually transmitted infections risk.**

| Reference | Population | n | STI risk (%) | Associated factors |
|---|---|---|---|---|
| *Kiene, Tennen & Armeli, (2008)* | Female college undergraduates who consumed alcohol and sexually active with casual partners | * | 25–30 | |
| *Ragsdale, Difranceisco & Pinkerton (2006)* | Female heterosexual tourists vacationing in a resort town, aged ≥18, sexually active on vacation and single (unaccompanied by a male 'romantic' partner) | 60 | 69 | No casual sex expectations, no alcohol consumption, being cautious regarding casual sex and embarrassed to discuss condoms |
| *Apostolopoulos, Sönmez & Yu (2002)* | Female undergraduates who had a sexual experience with someone they just met | 321 | 34.9 | |
| *Finney (2003)* | Second-year medical female students on holiday | 10 | 20 | Taking oral contraceptive pill |
| *De Visser et al. (2003)* | Heterosexual women who reported experience of vaginal or anal intercourse in the last year with casual partners | 324 | 63.8 | |
| *Hertliz (2009)* | Single women without a regular partner from the general population | 3,160 | 23 | |
| *Reece et al. (2010)* | Women aged 18–24 years old who their last vaginal intercourse was with casual partners | 78 | 69 | |
| *Paasche-Orlow et al. (2005)* | Incarcerated women | 423 | 61 | Low educational attainment |
| *Vandepitte et al. (2011)* | Women involved in commercial sex work | 1,027 | 40 | |
| *Berhan & Berhan (2012)* | Women aged 15–49 years | 207,776 | 17.6 | Living in urban areas, attained secondary and above education and owned middle to highest wealth index |

**Notes.**
*Not given.
 STI,  Sexually transmitted infections.

who attend universities (*Instituto Nacional de Estadística, 2005*). The main characteristics of this population are: age 18–25 years, middle to high socio-economic status, single status (in Spain the average age at first marriage is 31.68 years) (*Instituto Nacional de Estadística, 2012*), and interested in health sciences. Regarding sexual orientation, 2.7% of women report having had a homosexual relationship during their lifetime (*Instituto Nacional de Estadística, 2004*).

## Study design and participants

This cross-sectional observational study, undertaken between February 2005 and February 2009, selected a sample of university students studying healthcare sciences at Miguel Hernández University, Elche. The sample comprised all female third-year students studying medicine and second-year female students studying occupational therapy who attended lectures on a particular day during the study period and who wished to participate voluntarily. Data from women who did not wish to participate were not used for the analysis. The data were collected in classrooms with space for 100 to 200 students, but which are not generally full. In addition, all the participants had to have had vaginal intercourse with a man during the three months prior to completing the questionnaire and
not have a stable partner; i.e., have had at least one sexual encounter with someone she did not identify as a stable partner and this partner was open to sexual activities with others. This information was assessed by specific questions, shown in the Appendix of a previous paper (*Navarro-Cremades et al., 2015*).

## Variables and measurements

The information was collected with an original questionnaire (See Appendix in *Navarro-Cremades et al., 2015*). Prior to distributing the questionnaire, a verbal introduction was given explaining the voluntary and anonymous nature of the survey, as well as the confidentiality of all the data. The participants were requested to be sincere in their responses. The same researcher always provided the prior standard information, handed out the questionnaires, was present during their completion, and addressed any questions. The questionnaire took approximately 25 min to complete. The validity of the questionnaire was assessed during a prior pilot study that used 114 female third-year medicine undergraduates in May 2004. The results of this pilot study showed the good psychometric characteristics of the questionnaire (analysis of items and internal consistency, indexes of discrimination and factorial analysis) (*Van-der Hofstadt et al., 2007–2008*; *Navarro-Cremades et al., 2013*).

The questionnaire used in this study collected information about various female sexological aspects (*Van-der Hofstadt et al., 2007–2008*; *Navarro-Cremades et al., 2013*; *Navarro-Cremades et al., 2015*). This study only used those items considered most relevant by the research team. The main outcome variable was the risk of a STI. This was defined as not having used a male condom in at least one of their vaginal sexual relations with casual male partners. To assess this, questions were asked about contraception methods used (condoms, oral contraceptives, none, withdrawal, intrauterine device, vaginal ring, patch, several methods, or abstinence) (*Navarro-Cremades et al., 2015*). If a woman failed to use a condom in any act involving intercourse, she was considered to be positive for STI risk, as regardless of whether she used a condom in other sexual encounters or with other partners, she had a risk behaviour at least once, and could therefore have been infected. On the other hand, this study did not assess sexual relations with other males in women who had a stable partner.

The secondary variables analyzed included: sexual orientation (heterosexual, bisexual or other), how do you normally achieve orgasm? (no orgasm, several methods, or a single method) (in the questionnaire the possible answers were: during vaginal intercourse, through fantasies and daydreams, by stimulation from my partner, by self stimulation, through various of the previous methods, I don't have orgasms, by other methods. As the responses varied greatly, we formed the following groups: one method, several methods, none of the methods), desire to increase the frequency of sexual activities (yes and no), desire to have more variety in sexual activities (hour of the day, position, etc.) (yes and no), frequency of sexual intercourse with the partner (6 → 5–7 times/week; 5 → 3–4 times/week; 4 → 1–2 times/week; 3 → 2–3 times/month; 2 → once/month), and age (in years) (*Navarro-Cremades et al., 2015*).

## Sample size

The overall sample size comprised the 175 university students who completed the questionnaire. All had had vaginal intercourse with a man during the three months prior to completing the questionnaire and did not have a stable partner. To determine which women fulfilled these characteristics we analyzed all the questionnaires completed ($n = 565$) and assessed the questions *Sexual orientation, are you in a stable relationship*? and *Method of contraception that you use* (*Navarro-Cremades et al., 2015*). Thus, using a significance of 5% and an expected proportion of 38.1% (proportion of women <30 years of age in Spain who do not always use a male condom with casual partners), the expected error in the estimation of the STI risk was 7.2% (*Instituto Nacional de Estadística, 2004*).

## Statistical analysis

As the study involved data from different years, we first checked that there were no variations over time in any of the variables measured, using the chi-square test of Pearson or Fisher (qualitative data), and ANOVA or Kruskal-Wallis (quantitative data). In the event of a difference over time being found, time would then be added as an explanatory variable in all the statistical tests done; otherwise time would not be included in any analysis. After checking this possibility, absolute and relative frequencies were used to describe the qualitative variables and means plus standard deviations for the quantitative variables. A multivariate logistic regression model was used to estimate the adjusted odds ratios (ORs) in order to analyze the association between the STI risk and the study variables. As we had 52 events in our sample (women with a STI risk), we could only introduce 5 explanatory variables in the model (one for every 10 events). To select these variables, we constructed a stepwise model based on the likelihood ratio test. Using this method, we obtained the adjusted ORs for the selected combination of explanatory variables. The prognostic likelihood of the STI risk in the multivariate model was transformed into charts to help interpret the results. The likelihood ratio test and the Hosmer-Lemeshow test were carried out for the goodness-of-fit of the model. Furthermore, we represented the ROC curve for the predicted probabilities of STI risk given by the model. All analyses were performed at a 5% significance level and associated confidence intervals (CI) were estimated for each relevant parameter. All the analyses were performed using SPSS 19.

## Ethical considerations

This study was approved by the Ethics Commission of Miguel Hernández University, Elche (reference DMC.FNC.01.14). All the study participants agreed verbally to collaborate voluntarily, anonymously and freely, with no direct reward for their participation or penalization for non participation. Any woman who did not wish to participate could leave the classroom before the questionnaires were handed out. To ensure confidentiality no personal data were recorded that could lead to identification of the participants.

## RESULTS

A total of 601 female students attended class on the day in question, of whom nine declined the invitation to participate. Another 15 had not had any sexual activity during the previous

three months and did not, therefore, complete the questionnaire. Seven students did not hand in the questionnaire and five handed it in without filling it in, leaving a total of 565 completed questionnaires. Of these, 370 women were excluded because they either had a stable partner or because they had not indicated on the questionnaire the method of contraception used. On the other hand, 20 women were excluded because some of the selected variables for this study had missing values. Thus, the final sample comprised 175 women.

As no variable varied over time ($p > 0.05$), time was not used as an explanatory variable in any analysis. Table 2 summarizes the information concerning the analysis of the STI risk ($n = 175$). Most of the women who participated in the study were heterosexual (96.0%) and achieved orgasm through different methods (74.9%). Half were satisfied with the frequency of their sexual relations (52.6%) and one third desired more variety in their sex lives (34.3%). The mean age was almost 21 years (20.8), as the sample comprised university students.

The main contraceptive methods used were: male condom, 123 (70.3%); oral contraceptives, 14 (8.0%); contraceptive patches, 1 (0.6%); various, 8 (4.6%); none, 29 (19.0%). Thus, 52 of the 195 women surveyed did not always use a male condom (STI risk) (29.7%, 95% CI [22.9–36.5]%).

The factors significantly associated with the risk of STI ($p < 0.05/p < 0.025$ for the multiple comparison in Method of orgasms → Bonferroni correction) were: not having orgasms and not desiring an increased frequency of sexual relations. The model was significant and the Hosmer-Lemeshow test did not show discrepancies between expected and observed events. The area under the ROC curve was high (Fig. 1).

Figure 2 shows a box chart with the following elements: method of orgasm on the $x$-axis, with the $y$-axis showing the prognostic likelihood of the STI risk obtained from the multivariate logistic regression model. Figure 2 also shows that those women with no orgasm had a greater likelihood of a STI risk. The same process with the $x$-axis showing the desire for an increased frequency of sexual relations (Fig. 3) gave a greater likelihood of STI risk in the women who did not wish to increase the frequency of their sexual relations.

## DISCUSSION

In this study we found that one in every three women was at risk of contracting a STI by not using a male condom in all their casual vaginal sexual relations with men. A literature search of studies analyzing this problem found that it varied between 20% and 69%. The proportion of those at risk for a STI in our study was near the lower limit, indicating that this behaviour is less usual than in other populations. This may be because our participants were more aware of STI than the participants in other studies (Table 1).

Concerning the factors associated with the STI risk, the authors of these other studies found that no casual sex expectations, no alcohol consumption, being cautious regarding casual sex, embarrassed to discuss condoms, living in urban areas, with a middle to highest wealth index, and taking oral contraceptive pills were associated with a STI risk. On the other hand, there were discrepancies regarding educational attainment (Table 1). Regarding

**Table 2  Analysis of STI risk in female university students from Alicante (Spain). 2005–2009 data.**

| Variable | Total 175 $n(\%)/x \pm s$ | STI risk 52(29.7%) $n(\%)/x \pm s$ | Adj. OR | 95% CI | $p$-value |
|---|---|---|---|---|---|
| Sexual orientation: | | | | | |
| Heterosexual | 168(96.0) | 48(28.6) | N/M | N/M | N/M |
| Bisexual or other* | 7(4.0) | 4(57.1) | | | |
| Method of orgasm: | | | | | |
| No orgasm | 13(7.4) | 9(69.2) | 7.01 | 1.49–33.00 | 0.008[a] |
| Several methods | 131(74.9) | 32(24.4) | 0.77 | 0.31–1.90 | |
| A single method* | 31(17.7) | 11(35.5) | 1 | | |
| Desire to increase the frequency of sexual relations: | | | | | |
| Yes | 92(52.6) | 18(19.6) | 0.27 | 0.13–0.59 | <0.001 |
| No* | 83(47.4) | 34(41.0) | 1 | | |
| Desire to have more variety in sexual relations: | | | | | |
| Yes | 60(34.3) | 13(21.7) | 0.58 | 0.27–1.26 | 0.168 |
| No* | 115(65.7) | 39(33.9) | 1 | | |
| Age (years) | $20.8 \pm 2.2$ | $20.9 \pm 2.0$ | N/M | N/M | N/M |
| Frequency of sexual intercourse with the partner | $3.8 \pm 1.2$ | $4.0 \pm 1.4$ | 1.17 | 0.86–1.60 | 0.396 |

**Notes.**

STI, Sexually transmitted infections; Adj. OR, adjusted odds ratio; CI, Confidence interval; N/M, Not in the model.

*Reference.

[a]$p$-value for the complete factor. The $p$-values for the comparison with the reference are: (1) No orgasm: 0.014; (2) Several methods: 0.571. Frequency of sexual intercourse with partner (6 = 5–7 times/week; 5 = 3–4 times/week; 4 = 1–2 times/week; 3 = 2–3 times/month; 2 = once/month; 1 = Never). Goodness-of-fit of the model: (1) likelihood ratio test: $X^2 = 26.1$, $p < 0.001$; (2) Hosmer-Lemeshow test: $X^2 = 13.6$, $p = 0.092$.

the socioeconomic factors, our sample was taken from a population living in urban areas and with a high educational level. Maybe this could produce a higher prevalence of STI risk than the studies which found these factors associated with STI risk (*Paasche-Orlow et al., 2005*; *Berhan & Berhan, 2012*). In our study we only examined factors related with sexual behaviour, and cannot therefore compare the results obtained. Firstly, we found that those women who did not achieve orgasm had a greater STI risk. This behaviour may be due to the fact that these women attempt to achieve orgasms by not using a condom in order to experience greater contact during sexual intercourse. Although this factor has been found

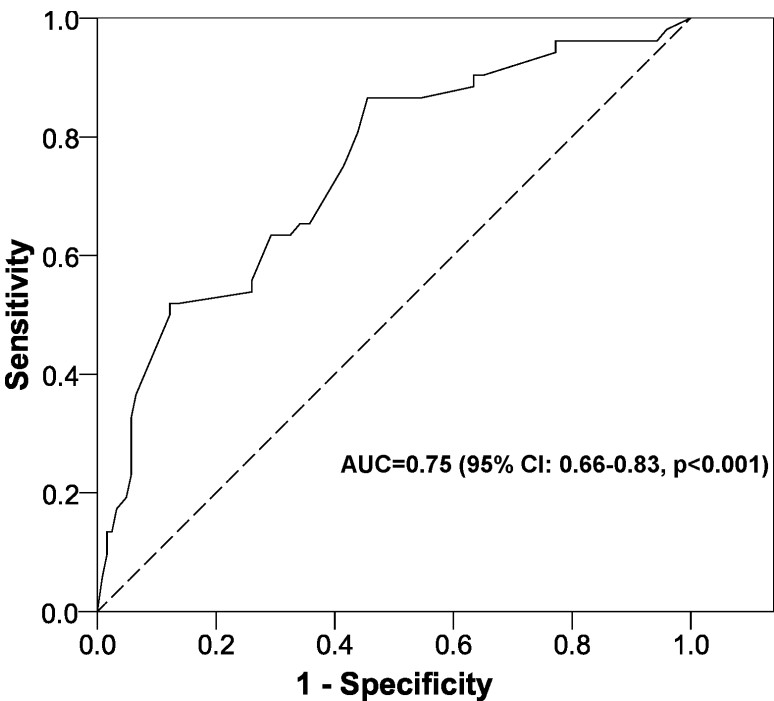

**Figure 1   ROC curve for the multivariate logistic regression model.** AUC, area under the ROC curve; CI, confidence interval.

to have an inverse association with infection by bacterial vaginosis, it is not comparable with the results of our study as, firstly the population of the other study was obtained from family planning clinics and secondly, we assessed the risk of contracting a STI whereas the other study determined the prevalence of the infection itself (*Nilsson et al., 1997*). Secondly, our results indicate that the women who were satisfied with the frequency of their sexual relations had a greater STI risk. This may be because persons who desire a greater sexual frequency acquire a more prudent behaviour regarding STI. Finally, although not quite reaching statistical significance, we found that less desire for greater sexual variety was also associated with this risk behaviour. As with the previous case, this may be due to the fact that persons who wish to increase their sexual variety are more aware of the risk, and therefore attempt to minimize the risk by greater use of a condom.

When we started this study we expected to find a lower magnitude of STI risk. However, the magnitude was unexpected, as one in every three women training to become a healthcare professional within a few years had a risk behaviour of contracting a STI. This is worrying, as in this population knowledge about STI and their prevention is higher than in the general population and yet STI risk behaviour was nevertheless very prevalent.

The fact that we found that not having orgasms was associated with the lack of condom use in this type of relations raises an important point often neglected in educational programs about STI prevention, as these women are putting their sexual pleasure before STI prevention. These results are important, considering that it is these very persons who should, in the future, make the population aware of the severity of this problem.

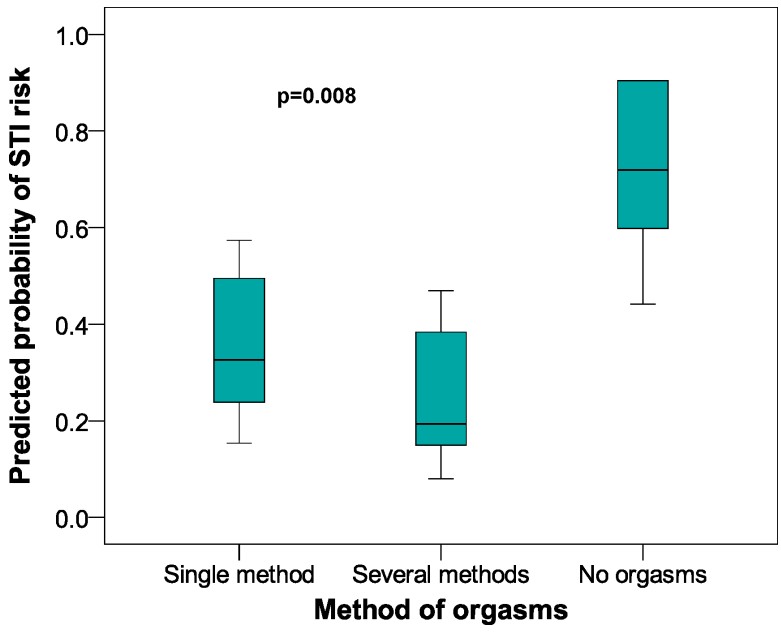

**Figure 2** **Predicted probabilities of STI risk in relation to Methods of Orgasm category in female university students from Alicante (Spain). 2005–2009 data.** STI, Sexually transmitted infections; CI, Confidence interval. In the questionnaire the possible answers for this item (how do you normally have orgasms?) were: during vaginal intercourse, through fantasies and daydreams, by stimulation from my partner, by self stimulation, through various of the previous methods, I don't have orgasms, by other methods. As the responses varied greatly, we formed the following groups: one method, several methods, none of the methods.

Healthcare personnel should be actively involved in the fight against STI by means of educational programs in the general population (*Fageeh, 2014*). Our results suggest that future members of this profession are not fully aware of the severity of the problem. Accordingly, the university curriculum should include specific programs about STI prevention and not rely solely on education, since it is clear that simply knowing that condoms are necessary to protect from STI infection is not nearly enough to modify behaviour. The hope is that when these students eventually become qualified healthcare personnel they can, in turn, raise awareness and promote motivation for protective and preventative methods amongst the general population. Another solution to this problem could be the condom use initiative and public awareness campaign, because other authors have shown that if the couple has used the condom in the past, the likelihood of using it in the future increases (*Shafii, Stovel & Holmes, 2007*). In other words, we could reduce the prevalence of the STI risk.

## Strengths and limitations of the study
The main strength of this study concerns the lack of studies analyzing the magnitude of this problem and its association with the sexual behaviour of women studying healthcare sciences. Thus, our results are innovative and indicate that sexual practices among these particular female university students are associated with the risk of contracting a STI.

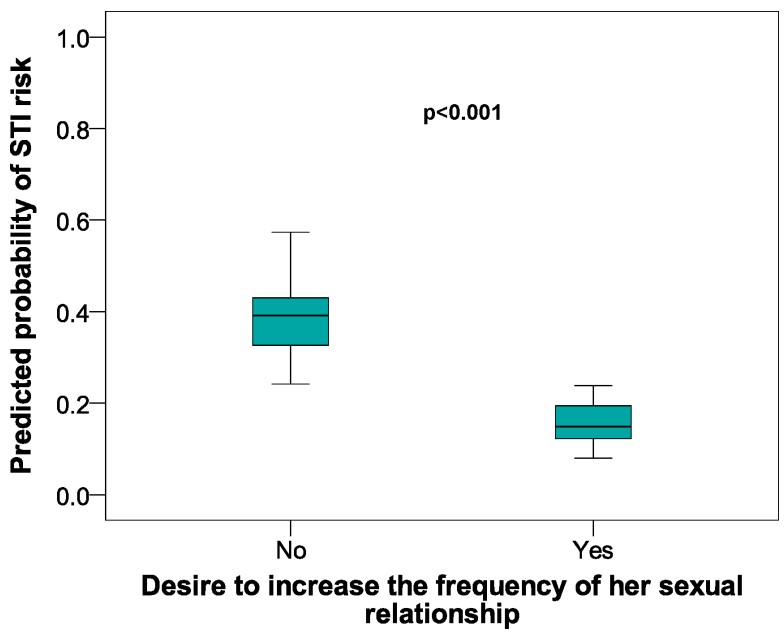

**Figure 3** **Predicted probabilities of STI risk in relation to Desire to increase the frequency of sexual re-lationships in female university students from Alicante (Spain). 2005–2009 data.** STI, Sexually transmit-ted infections; CI, Confidence interval.

The limitations are defined by its design. As this was a cross-sectional study we are unable to establish any temporality between the factors analyzed and the STI risk. This would require future longitudinal studies involving predictive models to determine which women are more likely to develop this particular behaviour and undertake an early intervention with educational activities to avoid the problem (STI risk). In order to minimize the information bias, the source corresponded to an instrument with good psychometric properties that reliably indicated the answers given by the participants (*Van-der Hofstadt et al., 2007–2008*; *Navarro-Cremades et al., 2013*). No measurements were made of factors that could influence the risk of STI, such as the attitude of the male partner, desire, arousal, lubrication, satisfaction, pain, sexual abuse, group sex practice, drugs and alcohol consumption, or the number of partners during the study period, though we did analyze the frequency of coital sexual activity during the study period. In addition, the AUC of the model was 0.75, so that the combination of our factors can explain the outcome satisfactorily. Also, the main outcome variable was the risk of STI and this was defined solely as not having used a male condom in at least one of their vaginal sexual relations with a casual male partner. To assess the real STI risk we have to take into account other aspects of the sexual encounters. This would increase the prevalence of STI risk. This issue will be considered in future studies. Finally, to minimize the selection bias, participants were selected once they had studied all the aspects of a STI. Thus, the students were aware of the lack of prevention and its consequences. We have to take into account that the objective of our study was to quantify the prevalence of STI in future health care personnel, not in the

general population, because we have to take measures at this stage to try to prevent STI in the general population, as discussed throughout the manuscript.

## CONCLUSIONS

Taking into account that these women had studied STI and their prevention, the mere knowledge did not appear to be enough to motivate behavioural changes. The fact that not having an orgasm was a risk factor for not using a condom suggests that people make decisions about condom use based on reinforcers such as pleasure rather than as the result of logical and analytical evaluation of the long-term pros and cons; therefore, our program should not just focus on education and providing knowledge, it needs to move towards increasing motivation to change (*Pollack, Boyer & Weinstein, 2013*). However, our results should be taken with caution because we have not analyzed other relevant factors for STI risk or determined the test–retest reliability (stability of responses over time). These issues will be studied in future research.

## ACKNOWLEDGEMENTS

We thank the Department of Applied Psychology of Miguel Hernández University, Elche, for allowing us to use the questionnaire for this study, and Felipe Navarro Sánchez for helping with the computerization of the data-base. The authors also thank Ian Johnstone for help with the English language version of the text. Finally, the authors thank Alessandra Rellini for her helpful comments.

### Funding

The authors received no funding for this work.

### Competing Interests

Antonio Palazón-Bru serves as an academic editor for PeerJ.

### Author Contributions

- Felipe Navarro-Cremades conceived and designed the experiments, contributed reagents/materials/analysis tools, wrote the paper, reviewed drafts of the paper.
- Antonio Palazón-Bru conceived and designed the experiments, analyzed the data, wrote the paper, prepared figures and/or tables, reviewed drafts of the paper.
- Dolores Marhuenda-Amorós, María Isabel Tomás-Rodríguez, Josefina Belda-Ibañez, Ángel Luis Montejo and Vicente Francisco Gil-Guillén conceived and designed the experiments, reviewed drafts of the paper.
- Fina Antón-Ruiz conceived and designed the experiments, contributed reagents/materials/analysis tools, reviewed drafts of the paper.

## Human Ethics

The following information was supplied relating to ethical approvals (i.e., approving body and any reference numbers):

This study was approved by the Ethics Commission of Miguel Hernández University, Elche (reference DMC.FNC.01.14). All the study participants agreed verbally to collaborate voluntarily, anonymously and freely, with no direct reward for their participation or penalization for non participation. Any woman who did not wish to participate could leave the classroom before the questionnaires were handed out. To ensure confidentiality no personal data were recorded that could lead to identification of the participants.

## Data Availability

Raw data has been uploaded to the Supplemental Information.

## Supplemental Information

Supplemental information for this article can be found online at http://dx.doi.org/10.7717/peerj.1699#supplemental-information.

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
