# Peer review of "Sexual behaviour and risk of sexually transmitted infections in young female healthcare students in Spain"

_PeerJ, doi:10.7717/peerj.1699_

## Round 0.1 · original submission · Major Revisions

Dear author,

Thank you for submitting your paper to PeerJ. As you see, the two reviewers have different opinions about the paper. Please go through the Reviewer no#2 comments and improve your manuscript accordingly before it can be considered further for publication.

Thanks
Ravi

·

Basic reporting

The present manuscript seeks to establish the role of women's sexual behaviour for STI risk. In this context, the finding that, women with no orgasm and who did not wish to increase the frequency of their sexual relations had a greater possibility of a STI risk is rather interesting and provides the basis for publication.

Experimental design

What is the probability of STI in women who had sexual intercourse without condom as well as with condom?

Validity of the findings

Following points should be considered for further improvement;

1. From Fig 1, does the p value (0.003) come from between 'No orgasms' and 'Single Method' or 'Several Methods'?
Since the error bars show that 'No orgasms' falls in the error ranges of both 'Single Method' and 'Several Methods', I'd like to know if they are statistically significant or not.
Similarly, from Fig 2, does the p value (p<0.001) come from between 'No' and 'Yes'?

2. For Fig 1, is there a possibility that women who don't feel orgasm tend to have their partners who don't like to use condoms? Since condom usage may be also dependent upon the will of male partners, this may be another factor to be considered.

Reviewer 2 ·

Basic reporting

Navarro-Cremades et al. describes sexual behavior and the risk of sexually transmitted infections in young female healthcare students in Spain.

The Introduction describes the scenario in Spain and some other parts of the globe, however, some references are not quoted.

In the background, the authors mention previous studies but none of them have focused specifically on sexual behavior and sexual function in women. Nevertheless, there are references in the literature that address risky sexual behavior. It was found that risky sexual behavior has a statistically significant association with women living in urban areas and with higher levels of education. In the case of HIV infection, higher education was often associated with a greater risk of infection. The author should discuss these findings in comparison with their results.

Regarding sexual function, previous findings have described higher orgasm ability in groups with different STIs. (Nilsson et al., 1997). How can the authors discuss this discrepancy?

The condom use initiative and public awareness campaign is not novel but it could be mentioned. The author did not quote previous findings related to the use of the condom, which is widely proven to prevent STI transmission (Shafii et al. 2007. Association between condom use, sexual behaviors and risk of sexually transmitted infections among adolescents.)

Experimental design

The authors claim the lack of information on risk of STI in relation with sexual behavior and sexual function in women. Nevertheless, the study population is very specific and composed of women studying healthcare.

In methods, the criteria used for the selection of the three groups is confusing. The authors divide the groups of participants with only 13 women in the group of “No orgasm”, 34 in the group of “single method” and 136 in the group of “Several methods.”

The experimental design in not completely appropriate for the research question. The main outcome variable was the risk of STI and it was defined solely as not having used a male condom in at least one of their vaginal sexual relations with a casual male partner.

To evaluate the risk of an STI, it would be relevant to assess other indicators such as sexual abuse, group sex practice, drugs and alcohol consumption. The questionnaire does not provide information about the number of sexual partners. A high number of lifetime sexual partners is considered a major indicator of high-risk sexual behavior and the most studied marker in measuring risk of acquiring a STI.

Any other domain in the aspect of female sexual function index such as, desire, arousal, lubrication, satisfaction and pain, was studied.

In the result section, the authors have shown that the factors associated significantly with the risk of STI were “not having
orgasms” and “not desiring an increased frequency of sexual relations.” They do not compare and discuss those finding with available information in the literature.

Validity of the findings

The manuscript does not show conclusive data and there are concerns regarding the technical approach.
Was the number of questionnaires completed 595 or 565?
The overall sample size was 195, however, sample size for the analysis for method of orgasm was 183.

The authors state that the result of the study suggests the need for future educational measures to prevent STI in healthcare students, however, the level of knowledge on the most common sexually transmitted infections and the risk of transmission was not evaluated in the questionnaire. How can the authors define the level of sexual education and prevention knowledge of the participants? Do they assume more sexual experience?

The questionnaire does not specify if the participants had received much information about STIs and HIV or if they would like to receive more information on these topics. The data can have implications for the conclusion and, it can indicate the need for sexual health education.

The conclusion tends to overvalue the findings, since the author pointed the orgasm only as an index for evaluation of sexual function without considering other domains (desire, arousal, lubrication, satisfaction and pain).
Also, for such funding it would be good to assess a test - retest reliability (stability of responses over time).

The author provided a speculative explanation for higher risk of STI in the group “No orgasm.” Previous findings have shown that women with orgasm ability are at higher risk of STI. It is important to discuss those results.

The author showed in table 2 that the desire to increase the frequency of her sexual relationship decreases the probability of STI risk. It should be widely discussed since the opposite is expected.

In the discussion section, the authors suggest that the lower proportion of the participants at risk for STI can be explained because their participants were more aware of STI than the participants from other studies. However, there is no evidence and we do not know their level of knowledge of the most common sexually transmitted infections. The grade knowledge on STI was not evaluated in the questionnaire.

Some related references are not quoted in the manuscript, Fageeh WM, BMC Infect Dis. 2014; Paasche-Orlow J Womens Health. 2005; Asres B, A meta-analysis on higher-risk sexual behavior of women, 2012; Pollack LM et al. Sex Transm Dis. 2013; Vandepitte J et al. Sex Transm Dis. 2011.

---

## Round 0.2 · accepted · Accept

Dear author,

Thank you for the revised manuscript. I am pleased to inform that your manuscript is suitable for the publication after the revision.

·

Basic reporting

Authors have answered and incorporated all the comments appropriately. Now, it is acceptable for the publication.

Experimental design

No comments.

Validity of the findings

No comments.